# Oracle-oriented Robustness: Robust Image Model Evaluation with Pretrained Models as Surrogate Oracle

## Abstract

Machine learning has demonstrated remarkable performances over finite datasets, yet whether the scores over the fixed benchmarks can sufficiently indicate the model's performances in the real world is still in discussion. In reality, an ideal robust model will probably behave similarly to the oracle (*e.g.*, the human users), thus a good evaluation protocol is probably to evaluate the models' behaviors in comparison to the oracle. In this paper, we introduce a new robustness measurement that directly measures the image classification model's performance compared with a surrogate oracle. Besides, we design a simple method that can accomplish the evaluation beyond the scope of the benchmarks. Our method extends the image datasets with new samples that are sufficiently perturbed to be distinct from the ones in the original sets, but are still bounded within the same causal structure the original test image represents, constrained by a surrogate oracle model pretrained with a large amount of samples. As a result, our new method will offer us a new way to evaluate the models' robustness performances, free of limitations of fixed benchmarks or constrained perturbations, although scoped by the power of the oracle. In addition to the evaluation results, we also leverage our generated data to understand the behaviors of the model and our new evaluation strategies.

## 1 Introduction

Machine learning has achieved remarkable performance over various benchmarks. For example, the recent successes of multiple pretrained models (Bommasani et al., 2021; Radford et al., 2021), with the power gained through billions of parameters and samples from the entire internet, has demonstrated human-parallel performance in understanding natural languages (Brown et al., 2020) or even arguably human-surpassing performance in understanding the connections between languages and images (Radford et al., 2021). Even within the scope of fixed benchmarks, machine learning has showed strong numerical evidence that the prediction accuracy over specific tasks can reach the position of the leaderboard as high as a human (Krizhevsky et al., 2012; He et al., 2015; Nangia & Bowman, 2019), suggesting multiple application scenarios of these methods.

However, these methods deployed in the real world often underdeliver its promises made through the benchmark datasets (Edwards, 2019; D'Amour et al., 2020), usually due to the fact that these benchmark datasets, typically *i.i.d*, cannot sufficiently represent the diversity of the samples a model will encounter after being deployed in practice.

Fortunately, multiple lines of study have aimed to embrace this challenge, and most of these works are proposing to further diversify the datasets used at the evaluation time. We notice these works mostly fall into two main categories: (1) the works that study the performances over testing datasets generated by predefined perturbation over the original *i.i.d* datasets, such as adversarial robustness (Szegedy et al., 2013; Goodfellow et al., 2015) or robustness against certain noises (Geirhos et al., 2019; Hendrycks & Dieterich, 2019; Wang et al., 2020b); and (2) the works that study the performances over testing datasets that are collected anew with a procedure/distribution different from the one for training sets, such as domain adaptation (Ben-David et al., 2007; 2010) and domain generalization (Muandet et al., 2013).

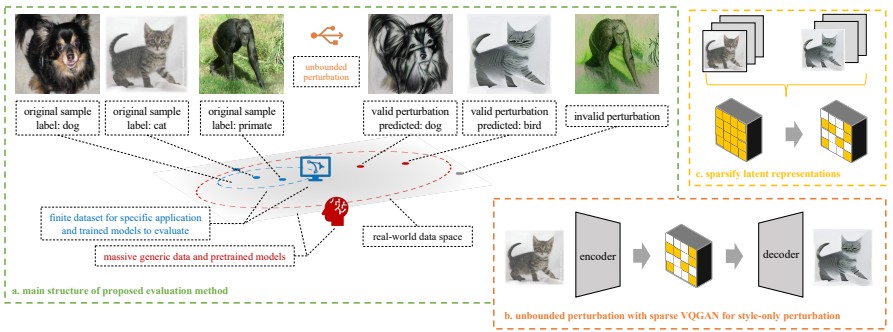

Figure 1: (a). the main structure of our system to generate test images with surrogate oracle and examples of the generated images with their effectiveness in evaluation of model's robustness. (b). the sparse VQGAN we used to introduce unbounded perturbation. (c). the sparse feature selection method to sparsify VQGAN.

Both of these lines, while pushing the study of robustness evaluation further, mostly have their own advantages and limitations as a tradeoff on how to guarantee the underlying causal structure of evaluation samples will be the same as the training samples: perturbation based evaluations usually maintain the causal structure by predefining the perturbations to be within a set of operations that will not alter the image semantics when applied, such as $\ell$-norm ball constraints (Carlini et al., 2019), or texture (Geirhos et al., 2019), frequency-based (Wang et al., 2020b) perturbations; on the other hand, new-dataset based evaluations can maintain the causal structure by soliciting the efforts to human annotators to construct datasets with the same semantics, but significantly different styles (Hendrycks et al., 2021b; Hendrycks & Dietterich, 2019; Wang et al., 2019; Gulrajani & Lopez-Paz, 2020; Koh et al., 2021; Ye et al., 2021). More details of these lines and their advantages and limitations and how our proposed evaluation protocol will contrast them will be discussed in the next section.

In this paper, we investigate how to diversify the robustness evaluation datasets to make the evaluation results credible and representative. As shown in Figure 1, we aim to integrate the advantages of the above two directions by introducing a new protocol to generate evaluation datasets that can automatically perturb the samples to be sufficiently different from existing test samples, while maintaining the underlying unknown causal structure with respect to an oracle (we use a CLIP model in this paper). Based on the new evaluation protocol, we introduce a new robustness measurement that directly measures the robustness compared with the oracle. With our proposed evaluation protocol and metric, we give a study of current robust machine learning techniques to identify the robustness gap between existing models and the oracle. This is particularly important if the goal of a research direction is to produce models that function reliably to have performance comparable to the oracle.

Therefore, our contributions in this paper are three-fold:

- We introduce a new robustness measurement that directly measures the robustness gap between models and the oracle.

- We introduce a new evaluation protocol to generate evaluation datasets that can automatically perturb the samples to be sufficiently different from existing test samples, while maintaining the underlying unknown causal structure.

- We leverage our evaluation metric and protocol to offer a study of current robustness research to identify the robustness gap between existing models and the oracle. Our findings further bring us understandings and conjectures of the behaviors of the deep learning models.

## 2 BACKGROUND

### 2.1 CURRENT ROBUSTNESS EVALUATION PROTOCOLS

The evaluation of machine learning models in non-*i.i.d* scenario have been studied for more than a decade, and one of the pioneers is probably *domain adaptation* (Ben-David et al., 2010). In

domain adaptation, the community trains the model over data from one distribution and test the model with samples from a different distribution; in *domain generalization* (Muandet et al., 2013), the community trains the model over data from several related distributions and test the model with samples from yet another distribution. To be more specific, a popular benchmark dataset used in domain generalization study is the PACS dataset (Li et al., 2017), which consists the images from seven labels and four different domains (photo, art, cartoon, and sketch), and the community studies the empirical performance of models when trained over three of the domains and tested over the remaining one. To facilitate the development of cross-domain robust image classification, the community has introduced several benchmarks, such as PACS (Li et al., 2017), ImageNet-A (Hendrycks et al., 2021b), ImageNet-C (Hendrycks & Dietterich, 2019), ImageNet-Sketch (Wang et al., 2019), and collective benchmarks integrating multiple datasets such as DomainBed (Gulrajani & Lopez-Paz, 2020), WILDS (Koh et al., 2021), and OOD Bench (Ye et al., 2021).

While these datasets clearly maintain the underlying causal structure of the images, a potential issue is that these evaluation datasets are fixed once collected. Thus, if the community relies on these fixed benchmarks repeatedly to rank methods, eventually the selected best method may not be a true reflection of the world, but a model that can fit certain datasets exceptionally well. This phenomenon has been discussed by several textbooks (Duda et al., 1973; Friedman et al., 2001). While recent efforts in evaluating collections of datasets (Gulrajani & Lopez-Paz, 2020; Koh et al., 2021; Ye et al., 2021) might alleviate the above potential hazards of "model selection with test set", a dynamic process of generating evaluation datasets will certainly further mitigate this issue.

On the other hand, one can also test the robustness of models by dynamically perturbing the existing datasets. For example, one can test the model's robustness against rotation (Marcos et al., 2016), texture (Geirhos et al., 2019), frequency-perturbed datasets (Wang et al., 2020b), or adversarial attacks (*e.g.*, $\ell_p$-norm constraint perturbations) (Szegedy et al., 2013). While these tests do not require additionally collected samples, these tests typically limit the perturbations to be relatively well-defined (*e.g.*, a texture-perturbed cat image still depicts a cat because the shape of the cat is preserved during the perturbation).

While this perturbation test strategy leads to datasets dynamically generated along the evaluation, it is usually limited by the variations of the perturbations allowed. For example, one may not be able to use some significant distortion of the images in case the object depicted may be deformed and the underlying causal structure of the images are distorted. More generally speaking, most of the current perturbation-based test protocols are scoped by the tradeoff that a minor perturbation might not introduce enough variations to the existing datasets, while a significant perturbation will potentially destroy the underlying causal structures.

## 2.2 ASSUMED DESIDERATA OF ROBUSTNESS EVALUATION PROTOCOL

As a reflection of the previous discussion, we attempt of offer a summary list of three desired properties of the datasets serving as the benchmarks for robustness evaluation:

- **Stableness in Causal Structure:** the most important property of the evaluation datasets is that the samples must represent the same underlying causal structure as the one in the training samples.
- **Diversity in Generated Samples:** for any other non-causal factors of the data, the test samples should cover as many as possible scenarios of the images, such as texture, styles *etc.*
- **A Dynamic Generation Process:** to mitigate selection bias of the models over techniques that focus too attentively to the specification of datasets, ideally, the evaluation protocol should consist of a dynamic set of samples, preferably generated with the tested model in consideration.

*Key Contribution:* To the best of our knowledge, there are no other evaluation protocols of model robustness that can meet the above three properties simultaneously. Thus, we aim to introduce a method that can evaluate model's robustness that fulfill the three above desiderata at the same time.

## 2.3 NECESSITY OF NEW ROBUSTNESS MEASUREMENT IN DYNAMIC EVALUATION PROTOCOL

In previous experiments, we always have two evaluation settings: the "standard" test set, and the perturbed test set. When comparing the robustness of two models, prior arts would be to rank the models by their accuracy under perturbed test set (Geirhos et al., 2019; Hendrycks et al., 2021a;

Orhan, 2019; Xie et al., 2020; Zhang, 2019) or other quantities distinct from accuracy, *e.g.,* inception score (Salimans et al., 2016), effective robustness (Taori et al., 2020) and relative robustness (Taori et al., 2020). These metrics are good starting points for experiments since they are precisely defined and easy to apply to evaluate robustness interventions. In the dynamic evaluation protocols, however, these quantities alone cannot provide a comprehensive measure of robustness, as two models are tested on two different "dynamical" test sets. When one model outperforms the other, we cannot distinguish whether one model is actually better than the other, or if the test set happened to be easier.

The core issue in the preceding example is that we can not find the consistent robustness measurement between two different test sets. In reality, an ideal robust model will probably behave similarly to the oracle (e.g., the human users). Thus, instead of indirectly comparing models' robustness with each other, a measurement that directly measures models' robustness compared with the oracle is desired.

## 3 METHOD - COUNTERFACTUAL GENERATION WITH SURROGATE ORACLE

### 3.1 METHOD OVERVIEW

We use $(\mathbf{x}, \mathbf{y})$ to denote an image sample and its corresponding label, use $\theta(\mathbf{x})$ to denote the model we aim to evaluate, which takes an input of the image and predicts the label.

We use $g(\mathbf{x}, \mathbf{b})$ to denote an image generation system, which takes an input of the starting image $\mathbf{x}$ to generate the another image $\widehat{\mathbf{x}}$ within the computation budget $\mathbf{b}$. The generation process is performed as an optimization process to maximize a scoring function $\alpha(\widehat{\mathbf{x}}, \mathbf{z})$ that evaluate the alignment between the generated image and generation goal $\mathbf{z}$ guiding the perturbation process. The higher the score is, the better the alignment is. Thus, the image generation process is formalized as

$$\widehat{\mathbf{x}} = \underset{\widehat{\mathbf{x}}=g(\mathbf{x},\mathbf{b}),\mathbf{b}<\mathbf{B}}{\arg\max} \alpha(g(\mathbf{x},\mathbf{b}),\mathbf{z}),$$

where $\mathbf{B}$ denotes the allowed computation budget for one sample. This budget will constrain the generated image not far from the starting image so that the generated one does not converge to a trivial solution that maximizes the scoring the function.

In addition, we choose the model classification loss $l(\theta(\widehat{\mathbf{x}}), \mathbf{y})$ as $\mathbf{z}$. Therefore, the scoring function essentially maximizes the loss of a given image in the direction of a different class.

**Algorithm 1** Counterfactual Image Generation with Surrogate Oracle

---

**Input:** $(\mathbf{X}, \mathbf{Y})$, $\theta$, $g$, $h$, total number of iterations $\mathbf{B}$
**Output:** generated dataset $(\widehat{\mathbf{X}}, \mathbf{Y})$
**for** each $(\mathbf{x}, \mathbf{y})$ in $(\mathbf{X}, \mathbf{Y})$ **do**
    generate $\widehat{\mathbf{x}}_0 = g(\mathbf{x}, \mathbf{b}_0)$
    **if** $h(\widehat{\mathbf{x}}_0) = \mathbf{y}$ **then**
        set $\widehat{\mathbf{x}} = \widehat{\mathbf{x}}_0$
        **for** iteration $\mathbf{b}_t < \mathbf{B}$ **do**
            generate $\widehat{\mathbf{x}}_t = g(\widehat{\mathbf{x}}_{t-1}, \mathbf{b}_t)$
            **if** $h(\widehat{\mathbf{x}}_t) = \mathbf{y}$ **then**
                set $\widehat{\mathbf{x}} = \widehat{\mathbf{x}}_t$
            **else**
                set $\widehat{\mathbf{x}} = \widehat{\mathbf{x}}_{t-1}$
                exit FOR loop
            **end if**
        **end for**
    **else**
        set $\widehat{\mathbf{x}} = \mathbf{x}$
    **end if**
    use $(\widehat{\mathbf{x}}, \mathbf{y})$ to construct $(\widehat{\mathbf{X}}, \mathbf{Y})$
**end for**

---

Finally, to maintain the unknown causal structure of the images, we leverage the power of the pretrained giant models to scope the generation process: the generated images must be considered within the same class by the pretrained model, denoted as $h(\widehat{\mathbf{x}})$, which takes in the input of the image and makes a prediction.

Connecting all the components above, the generation process will aim to optimize the following:

$$\widehat{\mathbf{x}} = \underset{\widehat{\mathbf{x}}=g(\mathbf{x},\mathbf{b}),\mathbf{b}<\mathbf{B},\mathbf{z}=l(\theta(\widehat{\mathbf{x}}),\mathbf{y})}{\arg\max} \alpha(g(\mathbf{x},\mathbf{b}),\mathbf{z}), \qquad \text{subject to} \quad h(\widehat{\mathbf{x}}) = \mathbf{y}.$$

Our method is generic and agnostic to the choices of the three major components, namely $\theta$, $g$, and $h$. For example, the $g$ component can vary from something as simple as basic transformations adding noises or rotating images to a sophisticated method to transfer the style of the images; on the other hand, the $h$ component can vary from an approach with high reliability and low efficiency such as

actually outsourcing the annotation process to human labors to the other polarity of simply assuming a large-scale pretrained model can function plausibly as a human.

In the next part, we will introduce our concrete choices of $g$ and $h$ leading to the later empirical results, which build upon the recent advances of vision research.

## 3.2 ENGINEERING SPECIFICATION

We use VQGAN (Esser et al., 2021) as the image generation system $g(\mathbf{x}, \mathbf{b})$, and the $g(\mathbf{x}, \mathbf{b})$ is boosted by the evaluated model $\theta(\mathbf{x})$ serving as the $\alpha(\widehat{\mathbf{x}}, \mathbf{z})$ to guide the generation process, where $\mathbf{z} = l(\theta(\widehat{\mathbf{x}}), \mathbf{y})$ is the model classification loss on current perturbed images.

The generation is an iterative process guided by the scoring function: at each iteration, the system add more style-wise transformations to the result of the previous iteration. Therefore, the total number of iterations allowed is denoted as the budget $\mathbf{B}$ (see Section 4.5 and Appendix H for details of finding the best perturbation). In practice, the value of budget $\mathbf{B}$ is set based on the resource concerns.

To guarantee the causal structure of images, we use a CLIP (Radford et al., 2021) model to serve as $h$, and design the text fragment input of CLIP to be *"an image of {class}"*. We directly optimize VQGAN encoder space which guided by our scoring function. We show the algorithm in Algorithm 1.

### 3.2.1 SPARSE SUBMODEL OF VQGAN FOR EFFICIENT PERTURBATION

While our method will function properly as described above, we notice that the generation process still have a potential limitation: the bound-free perturbation of VQGAN will sometimes perturb the semantics of the images, generating results that will be rejected by the oracle later and thus leading to a waste of computational efforts.

To counter this challenge, we use a sparse variable selection method to analyze the embedding dimensions of VQGAN to identify a subset of dimensions that are mainly responsible for the non-semantic variations.

In particular, with a dataset $(\mathbf{X}, \mathbf{Y})$ of $n$ samples, we first use VQGAN to generate a style-transferred dataset $(\mathbf{X}', \mathbf{Y})$. During the generation process, we preserve the latent representations of input samples after the VQGAN encoder in the original dataset. We also preserve the final latent representations before the VQGAN decoder that are quantized after the iterations in the style-transferred dataset. Then, we create a new dataset $(\mathbb{E}, \mathbf{L})$ of $2n$ samples, for each sample $(\mathbf{e}, l) \in (\mathbb{E}, \mathbf{L})$, $\mathbf{e}$ is the latent representation for the sample (from either the original dataset or the style-transferred one), and $l$ is labelled as 0 if the sample is from the original dataset and 1 if the style-transferred dataset.

Then, we train $\ell_1$ regularized logistic regression model to classify the samples of $(\mathbb{E}, \mathbf{L})$. With $\mathbf{w}$ denoting the weights of the model, we solve the following problem

$$\arg\min_{\mathbf{w}} \sum_{(\mathbf{e}, l) \in (\mathbb{E}, \mathbf{L})} l(\mathbf{e}\mathbf{w}, l) + \lambda \|\mathbf{w}\|_1,$$

and the sparse pattern (zeros or not) of $\mathbf{w}$ will inform us about which dimensions are for the style.

## 3.3 MEASURING ROBUSTNESS

**Oracle-oriented Robustness (OOR).** By design, the causal structures of counterfactual images will be maintained by the oracle. Thus, if a model has a smaller accuracy drop on the counterfactual images, it means that the model makes more similar predictions to oracle compared to a different model. To precisely define OOR, we introduce counterfactual accuracy (CA), the accuracy on the counterfactual images that our generative model successfully produces. As SA may influence CA to some extent, to disentangle CA from SA, we normalize CA with SA as OOR:

$$OOR = \frac{CA}{SA} \times 100\%$$

In settings where the oracle is human labors, OOR measures the robustness difference between the evaluated model and human perception. In our experiment setting, OOR measures the robustness difference between models trained on fixed datasets (the evaluated model) and the model trained on unfiltered, highly varied, and highly noisy data (the oracle CLIP model).

### 3.4 THE NECESSITY OF THE SURROGATE ORACLE

At last, we devote a short paragraph to reminder some readers that, despite the alluring idea of designing systems that forgo the usages of underlying causal structure or oracle, it has been proved or argued multiple times that it is impossible to create that knowledge with nothing but data, in either context of machine learning (Locatello et al., 2019; Mahajan et al., 2019; Wang et al., 2021) or causality (Bareinboim et al., 2020; Xia et al., 2021),(Pearl, 2009, Sec. 1.4).

## 4 EXPERIMENTS - EVALUATION AND UNDERSTANDING OF MODELS

### 4.1 EXPERIMENT SETUP

We consider four different scenarios, ranging from the basic benchmark MNIST (LeCun et al., 1998), through CIFAR10 (Krizhevsky et al., 2009), 9-class ImageNet (Santurkar et al., 2019), to full-fledged 1000-class ImageNet (Deng et al., 2009). For ImageNet, we resize all images to $224 \times 224$ px. We also center and re-scale the color values with $\mu_{RGB} = [0.485, 0.456, 0.406]$ and $\sigma = [0.229, 0.224, 0.225]$. The total number of iterations allowed (computation budget $\mathbf{B}$) of our evaluation protocol is set as 10. We conduct the experiments on a NVIDIA GeForce RTX 3090 GPU.

For each of the experiment, we report a set of four results:

- Standard Accuracy (SA): reported for references.

- Validation Rate (VR): the percentage of images validated by the oracle that maintains the causal structure.

- Oracle-oriented Robustness (OOR): the robustness of the model compared with the oracle.

### 4.2 ROBUSTNESS EVALUATION FOR STANDARD VISION MODELS

We consider a large range of models (Appedix J) and evaluate pre-trained variants of a LeNet architecture (LeCun et al., 1998) for the MNIST experiment and ResNet architecture (He et al., 2016a) for the remaining experiments. For ImageNet experiment, we also consider pretrained transformer variants of ViT (Dosovitskiy et al., 2020), Swin (Liu et al., 2021), Twins (Chu et al., 2021), Visformer (Chen et al., 2021) and DeiT (Touvron et al., 2021) from the *timm* library (Wightman, 2019). We evaluate the most recent ConvNeXt (Liu et al., 2022) as well. All models are trained on the ILSVRC2012 subset of IN comprised of 1.2 million images in the training and a total of 1000 classes (Deng et al., 2009; Russakovsky et al., 2015).

We report our results in Table 1. As expected, these models can barely maintain its performances when tested on data from different distributions, as shown by many previous works (*e.g.,* Geirhos et al., 2019; Hendrycks & Dietterich, 2019; Wang et al., 2020b).

Interestingly, on ImageNet, though both transformer-variants models and vanilla CNN-architecture model, *i.e.,* ResNet, attain similar clean image accuracy, transformer-variants substantially outperforms ResNet50 in terms of OOR under our dynamic evaluation protocol. We conjecture such performance gap is partly originated from the differences in training setups; more specifically, it may be resulted by the fact transformer-variants by default use strong data augmentation strategies while ResNet50 use none of them. The augmentation strategies (*e.g.,* Mixup (Zhang et al., 2017), Cutmix (Yun et al., 2019) and Random Erasing (Zhong et al., 2020), *etc.*) already naively introduce out-of-distribution (OOD) samples during training, therefore are potentially helpful for securing model

Table 1: The robustness test of standard models. We note 1) the VR of oracle is different for different datasets, but consistent in each dataset, 2) there exists performance gap between standard models and the oracle, and 3) transformer-variants outperforms vanilla ResNet in terms of OOR.

| Data | Model | SA | VR | OOR |
|---|---|---|---|---|
| MNIST | LeNet | 99.09 | 24.97 | 28.04 |
| CIFAR10 | ResNet18 | 95.38 | 62.80 | 54.88 |
| 9-class IN | ResNet18 | 92.30 | 81.63 | 30.28 |
| ImageNet | ResNet50 | 76.26 | 32.57 | 43.47 |
| | ViT | 82.40 | 32.57 | 50.55 |
| | DeiT | 78.57 | 32.58 | 55.05 |
| | Twins | 80.53 | 32.57 | 60.25 |
| | Visformer | 79.88 | 32.57 | 59.87 |
| | Swin | 81.67 | 32.58 | 69.73 |
| | ConvNeXt | 82.05 | 32.58 | 58.11 |

robustness towards data shifts. When equiping with the similar data augmentation strategies, CNN-architecture model, *i.e.,* ConvNext, has achieved comparable performance in terms of OOR. This hypothesis has also been verified in recent works (Bai et al., 2021; Wang et al., 2022). We will offer more discussions on the robustness enhancing methods in Section 4.3.

Besides comparing performance between different standard models, OOR brings us the chance to directly compare models with the oracle. Across all of our experiments, the OOR shows the significant gap between models and the oracle, which is trained on the unfiltered and highly varied data, seemingly suggesting that training with a more diverse dataset would help with robustness. This overarching trend has also been identified in (Taori et al., 2020). However, quantifying when and why training with more data helps is still an interesting open question.

We also notice that the VR tends to be different for different datasets. We conjecture this is due to how the oracle model understands the images and labels, more discussions is offered in Section 5.

## 4.3 ROBUSTNESS EVALUATION FOR ROBUST VISION MODELS

Recently, some techniques have been introduced to cope with corruptions or style shifts. For example, by adapting the batch normalization statistics with a limited number of samples (Schneider et al., 2020), the performance on stylized images (or corrupted images) can be significantly increased. Additionally, some more sophisticated techniques, *e.g.,* AugMix (Hendrycks et al., 2019), have also been widely employed by the community.

To investigate whether those OOD robust models can still maintain the performance under our dynamic evaluation protocol, we evaluate the pretrained ResNet50 models combining with the four leading methods from the ImageNet-C leaderboard, namely Stylized ImageNet training (SIN; (Geirhos et al., 2019)), adversarial noise training (ANT; (Rusak et al.)) as well as a combination of ANT and SIN (ANT+SIN; (Rusak et al.)), optimized data augmentation using Augmix (AugMix; (Hendrycks et al., 2019)), DeepAugment (DeepAug; (Hendrycks et al., 2021a)) and a combination of Augmix and DeepAugment (DeepAug+AM; (Hendrycks et al., 2021a)).

The results are displayed in Table 2. Surprisingly, we find that some common corruption robust models, *i.e.,* SIN, ANT, ANT+SIN, fail to maintain their power under our dynamic evaluation protocol. We take the SIN method as an example. The OOR of SIN method is 42.92, which is even lower than that of a vanilla ResNet50. As these methods are well fitted in the benchmark ImageNet-C, such results verify the weakness of relying on fixed benchmarks to rank methods. The selected best method may not be a true reflection of the real world, but a model well fit certain datasets, which in turn proves the necessity of our dynamic evaluation protocol.

Table 2: The robustness test of generated counterfactual images for OOD robust models. SA* represents the model's top-1 accuracy on ImageNet-C dataset. We note applying DeepAug+AM yields the best OOR under our dynamic evaluation protocol.

| Model | SA | SA$^*$ | VR | OOR |
|---|---|---|---|---|
| ResNet50 | 76.26 | 39.20 | 32.57 | 43.47 |
| ANT | 76.26 | 50.41 | 32.57 | 42.92 |
| SIN | 76.24 | 45.19 | 32.57 | 42.90 |
| ANT+SIN | 76.26 | 52.60 | 32.58 | 43.52 |
| DeepAug | 76.26 | 52.60 | 32.57 | 46.33 |
| Augmix | 76.73 | 48.31 | 32.57 | 53.36 |
| DeepAug+AM | 76.68 | 58.10 | 32.58 | 58.19 |

DeepAug, Augmix and DeepAug+AM perform better than SIN and ANT methods in terms of OOR as they are dynamically perturbing the datasets, which alleviates the hazards of "model selection with test set" to some extent. However, their performance is limited by the variations of the perturbations allowed, resulting in only a marginal improvement compared with the ResNet50 under our evaluation protocol.

In addition, we also visualize the counterfactual images generated according to the evaluated style-shift robust models in Figure 2. More results are shown in Appendix L. Specifically, we have the following observations:

**Preservation of local textual details.** A number of recent empirical findings point to an important role of object textures for CNN, where object textures are more important than global object shapes for CNN model to learn (Gatys et al., 2015; Ballester & Araujo, 2016; Gatys et al., 2017; Brendel & Bethge, 2019; Geirhos et al., 2019; Wang et al., 2020b). We notice our generated counterfactual images may preserve false local textual details, the evaluation task will become much harder since textures are no longer predictive, but instead a nuisance factor (as desired). For the counterfactual

image generated for the DeepAug method (Figure 2f), we produce a skin texture similar to chicken skin, and the fish head becomes more and more chicken-like. ResNet with DeepAug method is misled by this corruption.

**Generalization to shape perturbations.** Moreover, since our attack intensity could be dynamically altered based on the model's gradient while still maintaining the causal structures, the perturbation we produce would be sufficiently that not just limited to object textures, but even be a certain degree of shape perturbation. As it is acknowledged that networks with a higher shape bias are inherently more robust to many different image distortions and reach higher performance on classification and classification tasks, we observe that the counterfactual image generated for SIN (Figure 2b and Figure 2i) and ANT+SIN (Figure 2d and Figure 2k) methods are shape-perturbed and successfully attack the models.

**Recognition of model properties.** With the combination of different methods, the counterfactual images generated would be more comprehensive. For example, the counterfactual image generated for DeepAug+AM (Figure 2g) would preserve the chicken-like head of DeepAug's and skin patterns of Augmix's. As our evaluation method does not memorize the model it evaluated, this result reveals that our method could recognize the model properties, and automatically generate those hard counterfactual images to complete the evaluation.

Overall, these visualizations reveal that our dynamic evaluation protocol dynamically adjusts attack strategies based on different model properties, and automatically generates diversified counterfactual images that complements static benchmark, *i.e.,* ImageNet-C, to expose weaknesses for models.

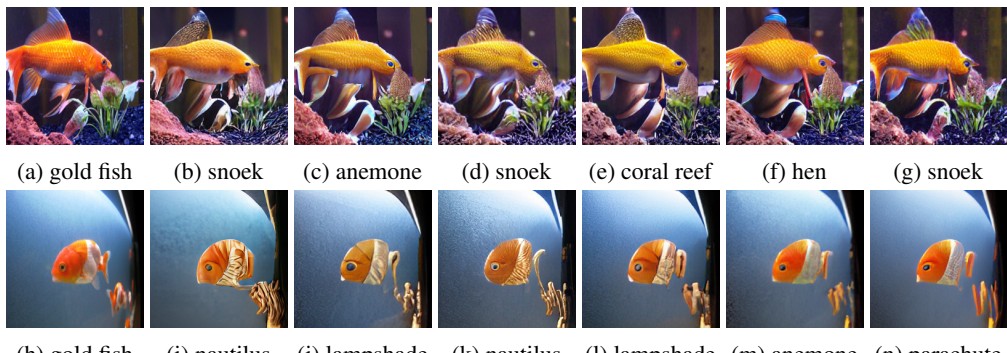

| (a) gold fish | (b) snoek | (c) anemone | (d) snoek | (e) coral reef | (f) hen | (g) snoek |

| (h) gold fish | (i) nautilus | (j) lampshade | (k) nautilus | (l) lampshade | (m) anemone | (n) parachute |

Figure 2: Visualization of the images generated by our system in evaluating the common corruption robust model, with the original image shown (left image of each row). The caption for each image is either the original label or the predicted label by the corresponding model. The evaluated models are SIN, ANT, ANT+SIN, Augmix, DeepAug and DeepAug+AM from left to right.

## 4.4 UNDERSTANDING THE PROPERTIES OF OUR EVALUATION SYSTEM

We continue to investigate several properties of the models in the next couple sections. To save space, we will mainly present the results on CIFAR10 experiment here and save the details to the appendix:

- In Appendix B, we explored the transferability of the generated images. The results of a reasonable transferability suggests that our method of generating images can be potentially used in a broader scope: we can also leverage the method to generate a static set of images and set a benchmark dataset to help the development of robustness methods.

- In Appendix C, we test whether initiating the perturbation process with an adversarial example will further degrade the OOR. We find that initiaing with the FGSM adversarial examples (Goodfellow et al., 2015) barely affect the OOR.

- In Appendix D, we compare the vanilla model to a model trained by PGD (Madry et al., 2017). We find that the adversarially trained model and vanillaly trained model process the data differently. However, their robustness weak spots are exposed to a similar degree by our test system.

- In Appendix E, we explored the possibility of improving the evaluated robustness by augmenting the images with the images generated by our evaluation system. However, due to the required

computational load, we only use a static set of generated images to train the model and the results suggest that static set of images for augmentation cannot sufficiently robustify the model to our evaluation system.

- We also notice that the generated images tend to shift the color of the original images, so we tested the robustness of grayscale models in Appendix F, the results suggest removing the color channel will not improve robustness performances.

## 4.5 Experiments Regarding Method Configuration

Table 3: Study of different image generator choices on ImageNet dataset. The numbers of VR and OOR are reported. The results of our dynamic evaluation protocol is consistent under different image generator configurations.

| Model | ADM | | Improved DDPM | | Efficient-VDVAE | | StyleGAN-XL | | VQGAN | |
|---|---|---|---|---|---|---|---|---|---|---|
| | VR | OOR | VR | OOR | VR | OOR | VR | OOR | VR | OOR |
| ResNet50 | 32.57 | 43.84 | 32.57 | 42.63 | 32.57 | 41.14 | 32.57 | 42.93 | 32.57 | 43.47 |
| ANT | 32.57 | 43.21 | 32.58 | 44.08 | 32.57 | 42.39 | 32.58 | 43.29 | 32.57 | 42.92 |
| SIN | 32.57 | 43.58 | 32.57 | 43.43 | 32.57 | 42.32 | 32.58 | 42.58 | 32.57 | 42.90 |
| ANT+SIN | 32.57 | 43.92 | 32.57 | 45.26 | 32.58 | 44.20 | 32.57 | 44.57 | 32.58 | 43.52 |
| DeepAug | 32.57 | 45.04 | 32.57 | 46.47 | 32.57 | 45.75 | 32.57 | 46.57 | 32.57 | 46.33 |
| Augmix | 32.58 | 52.77 | 32.57 | 53.69 | 32.58 | 53.42 | 32.57 | 52.57 | 32.57 | 53.36 |
| DeepAug+AM | 32.58 | 57.98 | 32.57 | 57.65 | 32.57 | 55.23 | 32.57 | 55.64 | 32.58 | 58.19 |

**Generator Configuration.** We conduct ablation study on the generator choice to agree on the performance ranking in Table 1 and Table 2. We consider several image generator architechitures, namely, variational autoencoder (VAE) (Kingma & Welling, 2013; Rezende et al., 2014) like Efficient-VDVAE (Hazami et al., 2022), diffusion models (Sohl-Dickstein et al., 2015) like Improved DDPM (Nichol & Dhariwal, 2021) and ADM (Dhariwal & Nichol, 2021), and GAN like StyleGAN-XL (Sauer et al., 2022). As shown in Table 3, the validation rate of the oracle stays stable across all the image generators. We find that the conclusion is consistent under different generator choices, which validates the correctness of our conclusions in Section 4.2 and Section 4.3.

**Sparse VQGAN.** In experiments of sparse VQGAN, we find that only 0.69% dimensions are highly correlated to the style. Therefore, we mask the rest 99.31% dimensions to create a sparse submodel of VQGAN for efficient perturbation. The running time can be reduced by 12.7% on 9-class ImageNet and 28.5% on ImageNet, respectively. Details can be found in Appendix G.

**Step size.** We experiment on the perturbation step size to find the best perturbation under the computation budget $\mathbf{B}$. We find that too small or large step size lead to slight perturbation strength while stronger image perturbation could be generated when the step size stays in a mild range, *i.e.,* 0.1 and 0.2. Details of our experiments on step size can be found in Appendix H.

## 5 Discussion and Conclusion

**Potential limitation.** We notice that, the CLIP model has been influenced by the imbalance sample distributions across the internet. We provide the details of test on 9-class ImageNet for vanilla ResNet-18 in Appendix I. We observe that the oracle model can tolerate a much more significant perturbation over samples labelled as *Dog* (VR 0.95) or *Cat* (VR 0.94) than samples labelled as *Primate* (VR 0.48). The OOR value for *Primate* images are much higher than other categories, creating an illusion that the evaluated models are robust against perturbed *Primate* images. However, such an illusion is caused by the limitation of the pretrained models that the oracle could only handle slightly perturbed samples.

**The usage of oracle.** *Is it cheating to use oracle?* The answer might depend on perspectives, but we hope to remind some readers that, in general, it is impossible to maintain the underlying causal structure during perturbation without prior knowledge (Locatello et al., 2019; Mahajan et al., 2019; Wang et al., 2021; Bareinboim et al., 2020; Xia et al., 2021),(Pearl, 2009, Sec. 1.4).

**Conclusion.** To conclude, in this paper, we first summarized the common practices of model evaluation strategies for robust vision machine learning. We then discussed three desiderata of the robustness evaluation protocol. Further, we offered a simple method that can fulfill these three desiderata at the same time, serving the purpose of evaluating vision models' robustness across generic *i.i.d* benchmarks, without requirement on the prior knowledge of the underlying causal structure depicted by the images, although relying on a plausible oracle.

ETHICS STATEMENT

The primary goal of this paper is to introduce a new evaluation protocol for vision machine learning research that can generate sufficiently perturbed samples from the original samples while maintaining the causal structures by assuming an oracle. Thus, we can introduce significant variations of the existing data while being free from additional human efforts. With our approach, we hope to renew the benchmarks for current robustness evaluation, offer understandings of the behaviors of deep vision models and potentially facilitate the generation of more truly robust models. Increasing the robustness of vision models can enhance their reliability and safety, which leads to the trustworthy artificial intelligence and contributes to a wide range of application scenarios (*e.g.,* manufacturing automation, surveillance systems, *etc.*). Manufacturing automation can improve the production efficiency, but may also trigger social issues related to job looses and industrial restructuring. Advanced surveillance systems are conducive to improving social security, but may also raise public concerns about personal privacy violations.

We encourage further work to understand the limitations of machine vision models in OOD settings. More robust models carry the potential risk of automation bias, i.e., an undue trust in vision models. However, even if models are robust against corruptions in finite OOD datasets, they might still quickly fail on the massive generic perturbations existing in the real-world data space, *i.e.,* the perturbations offered by our approach. Understanding under what conditions model decisions can be deemed reliable or not is still an open research question that deserves further attention.

REPRODUCIBILITY STATEMENT

Please refer to Appendix J for the references of all models we evaluated and links to the corresponding source code.

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

## A    NOTES ON THE EXPERIMENTAL SETUP

### A.1    NOTES ON MODELS

Note that we only re-evaluate existing model checkpoints, and hence do not perform any hyperparameter tuning for evaluated models. Since it is possible to work with a small amount of GPU resources, our model evaluations are done on a single NVIDIA GeForce RTX 3090 GPU.

### A.2    HYPERPARAMETER TUNING

Our method is generally parameter-free except for the computation budget and perturbation step size. In our experiments, the computation budget is the maximum iteration number of Sparse VQGAN. We consider the predefined value to be 10, as it guarantees the degree of perturbation with acceptable time costs. We provide the experiment for step size configuration in Section 4.5.

## B    TRANSFERABILITY OF GENERATED IMAGES

We first study whether our generated images are model specific, since the generation of the images involves the gradient of the original model. We train several architectures, namely EfficientNet (Tan & Le, 2019), MobileNet (Howard et al., 2017), SimpleDLA (Yu et al., 2018), VGG19 (Simonyan & Zisserman, 2014), PreActResNet (He et al., 2016b), GoogLeNet (Szegedy et al., 2015), and DenseNet121 (Huang et al., 2017) and test these models with the images. We also train another ResNet following the same procedure to check the transferability across different runs in one architecture.

Table 4 shows a reasonable transferability of the generated images as the OOR are all lower than the SA, although we can also observe an improvement over the OOR when tested in the new models. These results suggest that our method of generating images can be potentially used in a broader scope: we can also leverage the method to generate a static set of images and set a benchmark dataset to help the development of robustness methods.

Table 4: Performances of transferability.

| Model | SA | OOR |
|---|---|---|
| ResNet | 95.38 | 54.17 |
| EfficientNet | 91.37 | 68.48 |
| MobileNet | 91.63 | 68.72 |
| SimpleDLA | 92.25 | 66.16 |
| VGG | 93.54 | 70.57 |
| PreActResNet | 94.06 | 67.25 |
| ResNet | 94.67 | 66.23 |
| GoogLeNet | 95.06 | 66.68 |
| DenseNet | 95.26 | 66.43 |

In addition, our results might potentially help mitigate a debate on whether more accurate architectures are naturally more robust: on one hand, we have results showing that more accurate architectures indeed lead to better empirical performances on certain (usually fixed) robustness benchmarks (Rozsa et al., 2016; Hendrycks & Dietterich, 2019); while on the other hand, some counterpoints suggest the higher robustness numerical performances are only because these models capture more non-robust features that also happen exist in the fixed benchmarks (Tsipras et al., 2018; Wang et al., 2020b; Taori et al., 2020). Table 4 show some examples to support the latter argument: in particular, we notice that VGG, while ranked in the middle of the accuracy ladder, interestingly stands out when tested with generated images. These results continue to support our argument that a dynamic robustness test scenario can help reveal more properties of the model.

## C    INITIATING WITH ADVERSARIAL ATTACKED IMAGES

Since our method using the gradient of the evaluated model reminds readers about the gradient-based attack methods in adversarial robustness literature, we test whether initiating the perturbation process with an adversarial example will further degrade the accuracy.

We first generate the images with FGSM attack (Goodfellow et al., 2015). Table 5. shows that initiating with the FGSM adversarial examples barely affect the OOR, which

Table 5: Results on whether initiating with adversarial images ($\epsilon = 0.003$).

| Data | SA | OOR |
|---|---|---|
| regular | 95.38 | 57.80 |
| w. FGSM | 95.30 | 65.79 |

is probably because the major style-wise perturbation will erase the imperceptible perturbations the adversarial examples introduce.

## D   ADVERSARIALLY ROBUST MODELS

With evidence suggesting the adversarially robust models are considered more human perceptually aligned (Engstrom et al., 2019; Zhang & Zhu, 2019; Wang et al., 2020b), we compare the vanilla model to a model trained by PGD (Madry et al., 2017) ($\ell_\infty$ norm smaller than 0.03).

As shown in Table 6, adversarially trained model and vanillaly trained model indeed process the data differently: the transferability of the generated images between these two regimes can barely hold. In particular, the PGD model can almost maintain its performances when tested with the images generated by the vanilla model.

However, despite the differences, the PGD model's robustness weak spots are exposed to a similar degree with the vanilla model by our test system: the OOR of the vanilla model and the PGD model are only 57.79 and 66.18, respectively.

Table 6: Performances comparison with vanilla model and PGD trained model.

| Data | Model | SA | OOR |
|------|-------|------|------|
| Van. | Van. | 95.38 | 57.79 |
|      | PGD | 85.70 | 95.96 |
| PGD | Van. | 95.38 | 81.73 |
|      | PGD | 85.70 | 66.18 |

We believe this result can further help advocate our belief that the robustness test needs to be a dynamic process generating images conditioning on the model to test, and thus further help validate the importance of our contribution.

## E   AUGMENTATION THROUGH STATIC ADVERSARIAL TRAINING

Intuitively, inspired by the success of adversarial training (Madry et al., 2017) in defending models against adversarial attacks, a natural method to improve the empirical performances under our new test protocol is to augment the training data with counterfactual training images generated by the same process. We aim to validate the effectiveness of this method here.

However, the computational load of generation process is not ideal to serve the standard adversarial training strategy, and we can only have one copy of the counterfactual training samples. Fortunately, we notice that some recent advances in training with data augmentation can help learn robust representations with a limited scope of augmented samples (Wang et al., 2020a), which we use here.

We report our results in Table 7. The first thing we observe is that the model trained with the augmentation data offered through our approach could preserve a relatively higher performance (OOR 89.10) when testing with the counterfactual images generated according to the vanilla model. Since we have shown the counterfactual samples have a reasonable transferability in the main manuscript, this result indicates the robustness we brought when training with the counterfactual images generated by our approach.

Table 7: Test performances of the model trained in a vanilla manner (denoted as Van.) or with augmentation data offered through our approach (marked by the second column). We report two sets of performances, split by whether the counterfactual images are generated according to the vanilla model or the augmented model (marked by the first column).

| Data | Model | SA | OOR |
|------|-------|------|------|
| Van. | Van. | 95.38 | 57.79 |
|      | Aug | 87.41 | 89.10 |
| Aug. | Van. | 95.38 | 74.58 |
|      | Aug | 87.41 | 69.58 |

In addition, when tested with the counterfactual images generated according to the augmented model, both models' performance would drop significantly, which again indicates the effectiveness of our approach.

## F   GRAYSCALE MODELS

Our previous visualization suggests that a shortcut the counterfactual generation system can take is to significantly shift the color of the images, for which a grey-scale model should easily maintain the performance. Thus, we train a grayscale model by changing the ResNet input channel to be 1 and

transforming the input images to be grayscale upon feeding into the model. We report the results in Table 8.

Interestingly, we notice that the grayscale model cannot defend against the shift introduced by our system by ignoring the color information. On the contrary, it seems to encourage our system to generate more counterfactual images that can lower the performances.

Table 8: Test performances of the model trained in a vanilla manner (denoted as Van.) or with grayscale model. We report two sets of performances, split by whether the counterfactual images are generated according to the vanilla model or the grayscale one (marked by the first column).

| Data | Model | SA | OOR |
|------|-------|-------|-------|
| Van. | Van. | 95.38 | 57.79 |
|      | Gray | 93.52 | 66.06 |
| Gray | Van. | 95.38 | 67.48 |
|      | Gray | 93.52 | 44.76 |

In addition, we visualize some counterfactual images generated according to each model and show them in Figure 3. We can see some evidence that the grayscale model forces the generation system to focus more on the shape of the object and less of the color of the images. We find it particularly interesting that our system sometimes generates different images differently for different models while the resulting images deceive the respective model to make the same prediction.

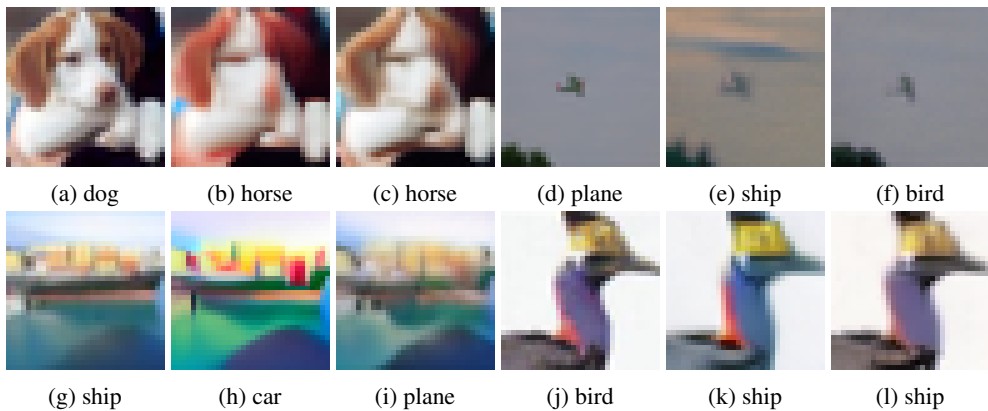

| (a) dog | (b) horse | (c) horse | (d) plane | (e) ship | (f) bird |

| (g) ship | (h) car | (i) plane | (j) bird | (k) ship | (l) ship |

Figure 3: Visualization of the counterfactual images generated by our system in evaluating the vanilla model (middle image of each group) and the grayscale model (third image of each group), with the original image shown. The caption for each image is either the original label or the predicted label by the corresponding model.

## G    EXPERIMENTS TO SUPPORT SPARSE VQGAN

We generate the flattened latent representations of input images after the VQGAN Encoder with negative labels. Following Algorithm 1, we generate the flattened final latent representations before the VQGAN decoder with positive labels. Altogether, we form a binary classification dataset where the number of positive and negative samples is balanced. The positive samples are the latent representations of counterfactual images while the negative samples are the latent representations of input images. We set the split ratio of train and test set to be $0.8 : 0.2$. We perform the explorations on various datasets, i.e. MNIST, CIFAR-10, 9-class ImageNet and ImageNet.

The classification model we consider is LASSO[1] as it enables automatically feature selection with strong interpretability. We set the regularization strength to be 36.36. We adopt saga (Defazio et al., 2014) as the solver to use in the optimization process. The classification results are shown in Table 9.

---

[1]Although LASSO is originally a regression model, we probabilize the regression values to get the final classification results.

Table 9: Classification results between vanilla and counterfactual images with LASSO.

| Data | Sparsity | Test score |
|---|---|---|
| MNIST | 97.99 | 78.50 |
| CIFAR-10 | 98.45 | 78.00 |
| 9-class ImageNet | 99.31 | 72.00 |
| ImageNet | 99.32 | 69.00 |

We observe that the coefficient matrix of features can be far sparser than we expect. We take the result of 9-class ImageNet as an example. Surprisingly, we find that almost 99.31% dimensions in average could be discarded when making judgements. We argue the preserved 0.69% dimensions are highly correlated to VQGAN perturbation. Therefore, we keep the corresponding 99.31% dimensions unchanged and only let the rest 0.69% dimensions participate in computation. Our computation loads could be significantly reduced while still maintain the competitive performance compared with the unmasked version[2].

We conduct the run-time experiments on a single NVIDIA GeForce RTX 3090 GPU. Following our experiment setting, we evaluate a vanilla ResNet-18 on 9-class ImageNet and a vanilla ResNet-50 on ImageNet. As shown in Table 10, the run-time on ImageNet can be reduced by 28.5% with our sparse VQGAN. Compared with large-scale masked dimensions (*i.e.,* 99.31%), we attribute the relatively incremental run-time improvement (*i.e.,* 12.7% on 9-class ImageNet, 28.5% on ImageNet) to the fact that we have to perform mask and unmask operations each time when calculating the model gradient, which offsets the calculation efficiency brought by the sparse VQGAN to a certain extent.

Table 10: Run-time Comparision between VQGAN and Sparse VQGAN.

| Method | Time | |
|---|---|---|
| | 9-class ImageNet | ImageNet |
| VQGAN | $521.5 \pm 1.2$s | $52602.4 \pm 2.7$s |
| Sparse VQGAN | $455.4 \pm 1.2$s | $40946.1 \pm 2.7$s |
| *Improv.* | 12.7% | 28.5% |

## H  PARAMETER STUDY ON STEP SIZE

We conduct the parameter study of the perturbation step size for our evaluation system on the CIFAR10 dataset. Specifically, we tune the step size in {0.01, 0.05, 0.1, 0,2, 0.5}. The maximum iteration (computation budget **B**) is set to be 10. All results are produced based on the ResNet18 and averaged over five runs.

As shown in Figure 4, we observe that when the step size is too small, *i.e.,* 0.01 and 0.05, the strength of perturbation cannot be achieved within the predefined maximum iterations, resulting in the higher score of OOR. In addition, large step size will also lead to higher OOR score. When the step size is large, *i.e.,* 0.5, the perturbation is likely to stop after only a few iterations. This could also lead to small perturbation strength compared with the scenario where we use relatively small step size but more iterations. When the step size is 0.01, the model seems achieves the oracle-parallel performance (OOR 99.66). However, such OOR values would become meaningless due to the small perturbation strength. Moreover, when the step size stays in a mild range, *i.e.,* 0.1 and 0.2, stronger image perturbation could be generated, while the performances at this range stay constant. Therefore, we choose the step size of 0.1 for the experiments.

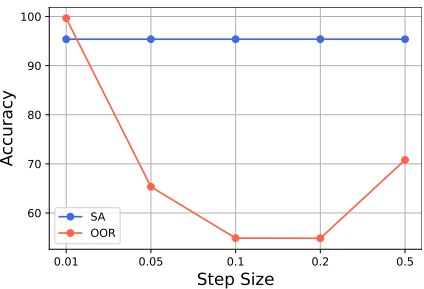

Figure 4: Study of different step sizes tested by ResNet.

---

[2]We note that the overlapping degree of the preserved dimensions for each dataset is not high, which means that we need to specify these dimensions when facing new datasets.

# I    ANALYSIS OF SAMPLES THAT ARE MISCLASSIFIED BY THE MODEL

We present the results on 9-class ImageNet experiment to show the details for each category.

Table 11: Details of test on 9-class ImageNet for vanilla ResNet-18

| Type | SA | VR | OOR |
|---|---|---|---|
| Dog | 93.33 | 95.33 | 17.98 |
| Cat | 96.67 | 94.00 | 31.55 |
| Frog | 85.33 | 80.67 | 20.34 |
| Turtle | 84.67 | 78.67 | 29.03 |
| Bird | 91.33 | 96.00 | 28.13 |
| Primate | 96.00 | 48.00 | 62.21 |
| Fish | 94.00 | 76.67 | 45.33 |
| Crab | 96.00 | 87.33 | 19.87 |
| Insect | 93.33 | 78.00 | 33.88 |
| Total | 92.30 | 81.63 | 30.28 |

Table 11 shows that the VR values for most categories are still higher than 80%, some even reach 95%, which means we produce sufficient number of counterfactual images. However, we notice that the VR value for *primate* images is quite lower compared with other categories, indicating around 52% perturbed *primate* images are blocked by the orcle. We have discussed this category unbalance issue in Section 5.

As shown in Table 11, the OOR value for each category significantly drops compared with the SA value, indicating the weakness of trained models. An interesting finding is that the OOR value for *Primate* images are quite higher than other categories, given the fact that more perturbed *Primate* images are blocked by the oracle. We attribute it to the limitation of foundation models. As the CLIP model has been influenced by the imbalance sample distributions across the Internet, it could only handle easy perturbed samples well. Therefore, the counterfactual images preserved would be those that can be easily classified by the models.

# J    LIST OF EVALUATED MODELS

The following lists contains all models we evaluated on various datasets with references and links to the corresponding source code.

## J.1    PRETRAINED VQGAN MODEL

We use the checkpoint of vqgan_imagenet_f16_16384 from `https://heibox. uni-heidelberg.de/d/a7530b09fed84f80a887/`

## J.2    PRETRAINED CLIP MODEL

Model weights of ViT-B/32 and usage code are taken from `https://github.com/openai/ CLIP`

## J.3    TIMM MODELS TRAINED ON IMAGENET (WIGHTMAN, 2019)

Weights are taken from `https://github.com/rwightman/pytorch-image-models/ tree/master/timm/models`

1. ResNet50 (He et al., 2016a)
2. ViT (Dosovitskiy et al., 2020)
3. DeiT (Touvron et al., 2021)
4. Twins (Chu et al., 2021)

5. Visformer (Chen et al., 2021)

6. Swin (Liu et al., 2021)

7. ConvNeXt (Liu et al., 2022)

## J.4 ROBUST RESNET50 MODELS

1. ResNet50 SIN+IN (Geirhos et al., 2019) `https://github.com/rgeirhos/texture-vs-shape`

2. ResNet50 ANT (Rusak et al.) `https://github.com/bethgelab/game-of-noise`

3. ResNet50 ANT+SIN (Rusak et al.) `https://github.com/bethgelab/game-of-noise`

4. ResNet50 Augmix (Hendrycks et al., 2019) `https://github.com/google-research/augmix`

5. ResNet50 DeepAugment (Hendrycks et al., 2021a) `https://github.com/hendrycks/imagenet-r`

6. ResNet50 DeepAugment+Augmix (Hendrycks et al., 2021a) `https://github.com/hendrycks/imagenet-r`

## J.5 ADDITIONAL IMAGE GENERATORS

1. Efficient-VDVAE (Hazami et al., 2022) `https://github.com/Rayhane-mamah/Efficient-VDVAE`

2. Improved DDPM (Nichol & Dhariwal, 2021) `https://github.com/open-mmlab/mmgeneration/tree/master/configs/improved_ddpm`

3. ADM (Dhariwal & Nichol, 2021) `https://github.com/openai/guided-diffusion`

4. StyleGAN (Sauer et al., 2022) `https://github.com/autonomousvision/stylegan_xl`

## K  LEADERBOARDS FOR ROBUST IMAGE MODEL

We launch leaderboards for robust image models. The goal of these leaderboards are as follows:

- To keep on track of state-of-the-art on each adversarial vision task and new model architectures with our dynamic evaluation process.
- To see the comparison of robust vision models at a glance (*e.g.,* performance, speed, size, *etc.*).
- To access their research papers and implementations on different frameworks.

We offer a sample of the robust ImageNet classification leaderboard in supplementary materials.

## L  ADDITIONAL COUNTERFACTUAL IMAGE SAMPLES

In Figure 5, we provide additional counterfactual images generated according to each model. We have similar observations to Section 4.3. First, the generated counterfactual images exhibit diversity that many other non-causal factors of the data would be covered, *i.e.,* texture, shape and styles. Second, our method could recognize the model properties, and automatically generate those hard counterfactual images to complete the evaluation.

In addition, the generated images show a reasonable transferability in Table 4, indicating tha our method can be potentially used in a broader scope: we can also leverage the method to generate a static set of images and set a benchmark dataset to help the development of robustness methods. Therefore, we also offer two static benchmarks in supplementary materials that are generated based on CNN architecture, *i.e.,* ConvNext and transformer variant, *i.e.,* ViT, respectively.

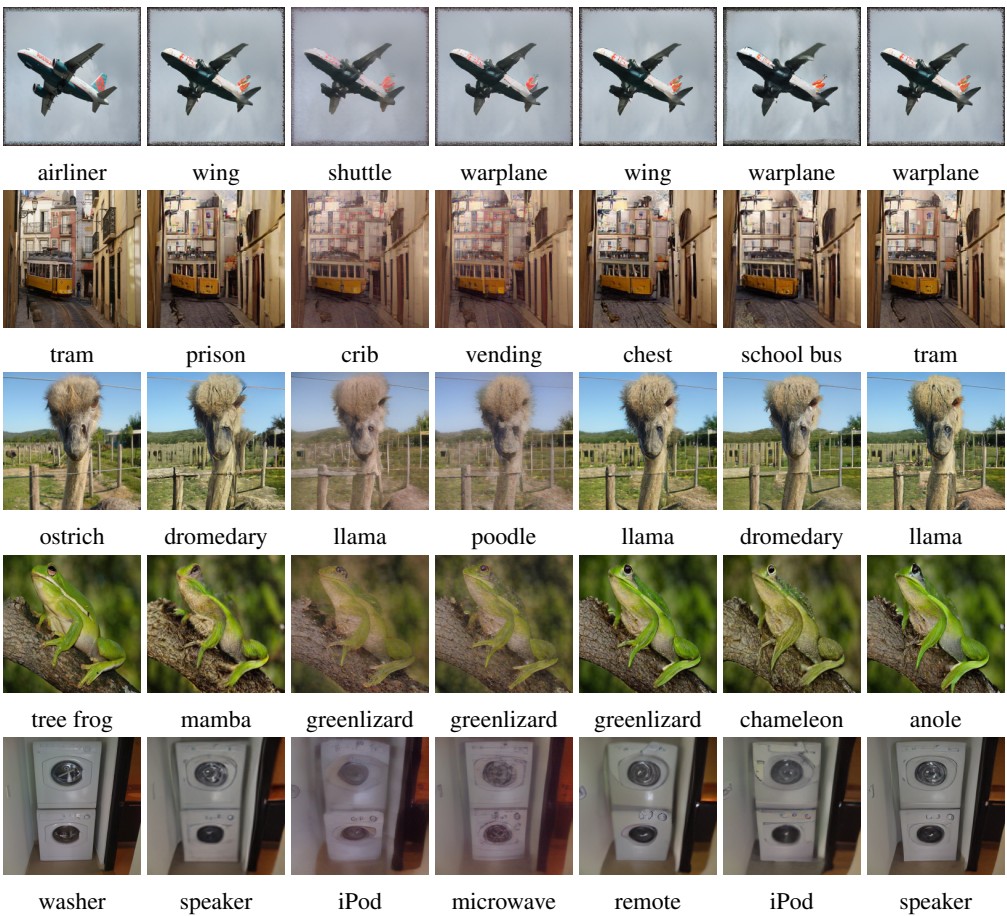

Figure 5: Visualization of the images generated by our system in evaluating the common corruption robust model, with the original image shown (left image of each row). The caption for each image is either the original label or the predicted label by the corresponding model. The evaluated models are SIN, ANT, ANT+SIN, Augmix, DeepAug and DeepAug+AM from left to right.

