# OpenReview forum: "Oracle-oriented Robustness: Robust Image Model Evaluation with Pretrained Models as Surrogate Oracle"
_ICLR.cc/2023/Conference — Submitted to ICLR 2023_

### Official Review · Reviewer_vhMp · 2022-10-20

**Confidence:** 4
**Correctness:** 2
**Technical Novelty And Significance:** 3
**Empirical Novelty And Significance:** Not applicable
**Recommendation:** 3

**Clarity, Quality, Novelty And Reproducibility:**

**Clarity**

Several key details are unclear or missing in Section 3.1.

* What exactly is the scoring function $\alpha$ ? Only the inputs to the functions are provided but the exact formualation is missing.
* What is the optimization objective with respect to the VQ-GAN latent space? How is the latent space perturbed?
* In Section 3.2.1, the paper describes a “style-transfer” process to sparsify the VQGAN latent space. Can the authors describe the process in detail?
* VR: Validation Rate: Why is the validation rate 100 or not close to 100, given that the algorithm introduces the constraint that the perturbed image has to be correctly classified in Section 3.1?
* I had to read until the methods section to understand that the oracle is a pre trained CLIP model. It would be nice if this is introduced as soon as possible in the introduction.




Some typos and rephrasing:

* as high as a human can reach -> as high as a human
* while push the study of robustness evaluation further -> while pushing the study
* More details of these lines
* oracle-parallel performance -> performance comparable to the oracle
* by assuming an oracle -> with respect to an oracle
* instead of indirectly compare models’ robustness -> instead of indirectly comparing models robustness
* in either machine learning context or causality context -> in either context of machine learning or causality.
* the accuracy on the images our generation process successfully produces a counterfactual image -> the accuracy on the counterfactual * images, that our generative model successfully produces
* s when tested by data from different distributions -> tested on data from different


**Strength And Weaknesses:**

Strengths:

The paper leverages large pretrained models to generate per-model out-of-distribution datasets dynamically which is new.

Weakness:

The paper is unclear on several details which are important to clarify before I can assess the experiments. Please see below.

**Summary Of The Paper:**

Several works stress test ImageNet classifiers on out-of-distribution benchmarks, which are static, defined either through pre-defined perturbations or a new test-set collection procedure. The paper proposes measuring out-of-distribution robustness through dynamically generated samples using VQGAN for each model instead, constrained to maintain semantic information . The oracle in this case is CLIP pretrained on a large number of text-image pairs.

**Summary Of The Review:**

See strengths and weaknesses above.

I think the paper could be of interest to the community but more work is required to provide all the adequate details necessary.

---

> ### Author Response · Authors · 2022-11-19
> **Response to Reviewer vhMp**
>
> We sincerely appreciate your kind comments and your positive assessment. We hope our point-to-point response can address your concerns.
>
> ### Regarding the clarity issues.
>
> **Details of the scoring function**
>
> We are sorry we missed the details of the scoring function, and thank you for pointing it out! The scoring function essentially maximizes the classification loss of the evaluated model to guide the generation of images. We have incorporated the details in the updated draft. Please kindly check the blue text in the method section.
>
> **Details of the perturbation of latent space**
>
> We are sorry we missed the details of the guidance process, and thank you for pointing it out! We directly optimize VQGAN encoder space, which is similar with CLIP-guided VQGAN, yet there are some key differences. We do not need the prompt representations encoded by the CLIP. The whole generation process is guided by the classification loss of the evaluated model. We have updated the details in the revised version. We hope these materials may ease your concern about the guidance process.
>
> **Details of the sparsification of VQGAN.**
>
> We are sorry for the confusion. We begin with the generation of a style-transferred dataset based on the original dataset and vanilla VQGAN. To find out the feature dimensions related to style, we utilize LASSO to classify the original images and style-transferred ones and obtain the sparse coefficient matrix of features that LASSO uses when making judgments. We kindly argue the preserved dimensions are highly correlated to VQGAN perturbation and regard them as the style dimensions. Therefore, we only let the identified style-related dimensions participate in computation. Our computation loads could be significantly reduced while still maintain the competitive performance compared with the unmasked version. Please kindly check Appendix G for more details.
>
> **Regarding the validation rate**
>
> We are sorry for the confusion. We attribute this phenomenon to the limited computing budget and relatively large perturbation step size. We conduct our experiment on an NVIDIA GeForce RTX 3090 GPU. With the limited computing resources, we have to set a relatively large perturbation step size in order to reach a tradeoff between image quality and efficiency. In this case, the images are harder to maintain their causal structures at the beginning due to the large perturbation step size, leading to the abnormal phenomenon of VR. If we have more computing resources, we can use a smaller perturbation step size to obtain fine-grained counterfactual images, and observe the corresponding phenomenon reflected in VR.
>
> **Regarding the clarification of the oracle**
>
> We are sorry for the confusion and thank you for pointing it out! We have updated it in our revised version.
>
> ### Revision of paper
>
> We sincerely appreciate your suggestions and fixed the typos and rephrasing in the revised paper.

---

### Official Review · Reviewer_vtR9 · 2022-10-21

**Confidence:** 5
**Correctness:** 3
**Technical Novelty And Significance:** 2
**Empirical Novelty And Significance:** 2
**Recommendation:** 3

**Clarity, Quality, Novelty And Reproducibility:**

- I feel that the quality of this paper has a large room for improvement. For example, this paper borrows some terminologies from the field of causality (e.g., "causal structure", "counterfactual", "confounder"), but they are not well-defined in this paper. Also, I feel this paper is not self-contained. For example, there is no detailed explanation of how the generative model is guided by classification loss. I also recommend making Section 4.4. self-contained in the main text.
- In terms of novelty and originality, I think this paper has limited novelty and contribution as my previous comment.

**Strength And Weaknesses:**

## Strength

- This paper tackles a very important problem in the robustness benchmark: we need a dynamic benchmark (based on optimization) rather than a manually collected dataset. Also, the generated dataset should be realistic (not based on l2 or l-infinity ball).
- This paper proposes a sparse submodel of VQGAN for reducing the computational cost

## Weakness

- I feel this paper has a limited novelty. The main modules are from the other works (VQGAN, CLIP). The idea of "adversarial attack by generative model" is an old and popular idea [R1-5]. The novel part of this paper could be (1) using a novel generative model rather than GAN, and (2) filtering based on the CLIP model, but I think these contributions are very limited. Furthermore, using VQGAN and CLIP models could be problematic as my next comment.
    - [R1] Song, Yang, et al. "Constructing unrestricted adversarial examples with generative models." Advances in Neural Information Processing Systems 31 (2018).
    - [R2] Xiao, Chaowei, et al. "Generating adversarial examples with adversarial networks." arXiv preprint arXiv:1801.02610 (2018).
    - [R3] Poursaeed, Omid, et al. "Generative adversarial perturbations." Proceedings of the IEEE Conference on Computer Vision and Pattern Recognition. 2018.
    - [R4] Qiu, Haonan, et al. "Semanticadv: Generating adversarial examples via attribute-conditioned image editing." European Conference on Computer Vision. Springer, Cham, 2020.
    - [R5] Jang, Yunseok, et al. "Adversarial defense via learning to generate diverse attacks." Proceedings of the IEEE/CVF International Conference on Computer Vision. 2019.
- I feel that the quality of the proposed generation process is still limited. Furthermore, I feel that the proposed method is highly biased toward VQGAN and CLIP, leading to biased evaluation results.
    - The proposed method highly depends on extra modules, such as VQGAN and CLIP. Namely, the quality or diversity of the generated images will be bounded by the VQGAN performance. Similarly, the quality of the generated images is bounded by the CLIP zero-shot generalizability to the generated images; if CLIP zero-shot performs worse for specific types of perturbation, then the proposed framework cannot cover such perturbation. As shown in the "Potential limitation" paragraph, the CLIP zero-shot performance is not consistent across the classes. It could cause a biased benchmark toward the selected models (i.e., CLIP).
    - As shown in Table 1-2, only about one-third of images are changed by VQGAN for ImageNet. It means that VQGAN and CLIP models cannot generate a proper image, or classify the generated images correctly. I think the quality of the proposed generation process is still not effective for measuring robustness.
    - Not only the generated samples will be biased toward CLIP, but also the metric will be biased toward CLIP as well. Because the generated samples are biased to the CLIP zero-shot performance, "CA" score will also be biased.
- Also, this benchmark is only available when a strong and generalizable generative model and a strong and generalizable "surrogated oracle" models are available. Thus, this benchmark is only limited to natural image benchmarks, such as ImageNet. This is not a strong weakness, but it is worth to be discussed.

## Questions and minor comments

- There is no detailed explanation of how VQGAN is guided by classification loss. I presume that the guidance for VQGAN is done by directly optimizing VQGAN encoder space (as the CLIP-guided VQGAN), but it would be great if the authors will provide more details for this.
- I feel that the concepts of "counterfactual" and "confounder" are used incorrectly in Section 3.3.
    - What does "counterfactual" mean here? For example, in the field of causality, the terminology "counterfactual explanation" describes the causal relationship by "If X has not occurred then Y does not occur". Similarly, in machine learning, a counterfactual example means a modified example with the minimum changes (to see what affects the prediction at most). However, the terminology "counterfactual accuracy" is weird. How accuracy becomes "counterfactual"?
    - Why the standard accuracy acts as a confounder of "counterfactual accuracy"? How standard accuracy and counterfactual accuracy are related in terms of causality? I cannot find any relationship between them, as well as, it is not trivial to treat "accuracies" as random variables. Also, to define a confounder (as far as the reviewer understood) we need to define "Do operation". What is "do operation" for accuracy scores?
- The formulation for $\hat x$ seems weird. I recommend avoiding using the same notation for the optimized results ($\hat x$ in LHS) variable ($\hat x$ in RHS, below $\arg\max$).
- Some citations are missing. For example, Section 4.2. missed citations for Mixup, CutMix, and random erasing (Cutout and RandAug are worth being cited as well).

**Summary Of The Paper:**

This paper proposes an adversarial attack benchmark where the attacked samples are generated by high-quality generative models filtered by a "surrogated oracle" (a model trained with large-scale extra data points, such as CLIP). More specifically, the proposed method generates adversarial samples by maximizing classification loss (i.e., cross-entropy loss) using VQGAN. The generated image is validated by the CLIP model, and if the CLIP model predicts the generated image with a wrong label, then the generated image is set to the original image. With the generated adversarial samples, this paper shows that the existing vision models are not robust to the proposed attack method.

**Summary Of The Review:**

- I think this paper has very limited novelty and the quality of this paper has a large room for improvement.
- Furthermore, the proposed method highly depends on VQGAN and CLIP models. I think it will cause serious problems: (1) the generation quality itself is limited by VQGAN and CLIP performances (2) the metric is biased toward the CLIP zero-shot performance. Hence, measuring the proposed benchmark will measure how a model is similar to the CLIP model, not measure how a model is robust.
- Overall, I recommend "reject".

---

Post-rebuttal comment: As my last comment, my concerns are somewhat addressed by the response, but I feel that my main concerns still remain after reading the response, the other reviews, and the revised paper. Hence, I will maintain my initial recommentation.

---

> ### Author Response · Authors · 2022-11-19
> **Response to Reviewer vtR9**
>
> We sincerely appreciate your kind comments and your insightful suggestions. Regarding the weaknesses, we hope our point-to-point response can address your concerns.
>
> **W1. Regarding previous studies on adversarial attack by generative models**
>
> Thank you for pointing out these papers! Indeed, the papers also propose generative methods for adversarial attack, yet there are some key differences between these papers and ours. Specifically, these papers study the standard adversarial attack and defence problem. By contrast, we focus on the robustness evaluation problem where we incorporate the oracle to defend the causal structure.
>
> **W2.1 Bias towards the generator**
>
> Yes, we agree with you that it is inevitable to introduce extra bias when we select specific generators and oracles. We kindly argue that we can alleviate such selection bias to some extent with our dynamic generation process. Meanwhile, the introduced bias is acceptable given the results of different generator choices shown in Table 3, where different generators produce consistent results, and there is no obvious bias towards a specific generator.
>
> **W2.2 Regarding the low VR value & bias towards the oracle**
>
> This is a good point, and thank you for pointing it out! We attribute this phenomenon to the limited computing budget and relatively large perturbation step size. We conduct our experiment on an NVIDIA GeForce RTX 3090 GPU. With the limited computing resources, we have to set a relatively large perturbation step size in order to reach a tradeoff between image quality and efficiency. In this case, the images are harder to maintain their causal structures at the beginning due to the large perturbation step size, leading to relatively small VR and insufficient perturbed images. With sufficient computing resources, we can set a relatively smaller stepsize to perturb the image little by little and can get enough more perturbed images. Admittedly, the oracle's bias still exists here, e.g., the chimpanzee images are still easier than other ones. However, considering the huge performance gap between oracle and the evaluated models, images that are easy for oracle are hard enough for the evaluated models, which is enough for the applications in reality. We leave the deployment on sufficient computing resources as a future work.
>
> **W2.3 Metrics are biased towards the oracle**
>
> Yes. By design, OOR measures the robustness difference between models trained on fixed datasets (the evaluated model) and the model trained on unfiltered, highly varied, and highly noisy data (the oracle CLIP model). Therefore, it is defined based on the oracle.
>
> **W3. Regarding the availability of strong surrogate oracles**
>
> Yes. The benchmark is available when a strong and generalizable generative model and a strong and generalizable "surrogated oracle" models are available. Otherwise, we have to find new ways to maintain the causal structure.
>
> **W4. Regarding the questions**
>
> Q1. Details of the guidance of VQGAN.
>
> A1. We are sorry we missed the details of the guidance process, and thank you for pointing it out! Regarding the suggestions, your understanding of the guidance process is correct. We directly optimize VQGAN encoder space, which is similar with CLIP-guided VQGAN, yet there are some key differences. We do not need the prompt representations encoded by the CLIP. The whole generation process is guided by the classification loss of the evaluated model. We have updated the details in the revised version. We hope these materials may ease your concern about the guidance process.
>
> Q2.1. Regarding the counterfactual accuracy
>
> A2.1. We are sorry about the confusion, and thank you for pointing it out! The terminology "counterfactual accuracy" means the models' classification accuracy on the counterfactual images, not the accuracy itself becomes counterfactual.
>
> Q2.2. Relationship between CA and SA.
>
> A2.2. The standard accuracy (SA) will influence the counterfactual accuracy (CA). When the CA is low, it may be that the model is not robust enough on the counterfactual images, but it may also be that the performance of the model itself (SA) is not good. Therefore, when comparing robustness of two models on counterfactual images, we need to disentangle the robustness of a model from its accuracy on the standard test set (SA). Despite the above reasons, the term "confounder" used here is not suitable. We have updated our expression in the revised version. Please kindly check the blue text in Section 3.3.
>
> **Revision of paper**
>
> We sincerely appreciate your suggestions and fixed the typos and citations in the revised paper.

---

> > ### Comment · Reviewer_vtR9 · 2022-11-21
> > **Response**
> >
> > Thanks for your comment, and sorry for the late reply. It partially addressed my concerns and questions. For example, I appreciate the authors for clarifying how the VQGAN guidance is employed in the revised paper.
> >
> > Although the response partially addressed my concerns, my main concerns still remained. I recap my previous comment:
> >
> > > - I think this paper has very limited novelty and the quality of this paper has a large room for improvement.
> > > - Furthermore, the proposed method highly depends on VQGAN and CLIP models. I think it will cause serious problems: (1) the generation quality itself is limited by VQGAN and CLIP performances (2) the metric is biased toward the CLIP zero-shot performance. Hence, measuring the proposed benchmark will measure how a model is similar to the CLIP model, not measure how a model is robust.
> >
> > I raised four issues (1) limited novelty (2) quality of the paper (3) generation quality is limited by VQGAN and CLIP (4) the metric is biased toward CLIP.
> >
> > I don't feel that the revision is enough to enhance the quality, but it is my fault that my response is after the revision deadline. Hence, in this final comment, I will lower the weight for (2) for a fair recommendation.
> >
> > However, as the response, the metric and method are still biased toward CLIP and VQ-GAN. I agree with that Table 3 could show that the proposed method is robust to the choice of generator. However, as the response, the metric itself is still biased toward CLIP, and the proposed method shows a low validation rate. The authors may assume that it is because of the computational resource, but if we would like to argue that, I would like to expect a small-scale experiment for measuring the relationship between computational resources and VR.
> >
> > In terms of novelty, the method is highly dependent on CLIP and VQ-GAN, especially CLIP. The VQ-GAN guidance itself is not a novel technical contribution. Also, the response argues that the proposed method is novel because my listed papers are based on adversarial defense, but the proposed method focuses on a robust evaluation. However, even if we move the focus of this paper to (non-adversarial) robustness benchmarks, there are some works applying the adversarial training scheme for robustness, e.g., ANT (Rusak et al.). Moreover, I don't think applying adversarial training to non-adversarial benchmarks itself is not a novel contribution.
> >
> > Overall, my concerns are somewhat addressed by the response, but I feel that my main concerns still remain after reading the response, the other reviews, and the revised paper.

---

### Official Review · Reviewer_3rws · 2022-10-24

**Confidence:** 4
**Correctness:** 3
**Technical Novelty And Significance:** 4
**Empirical Novelty And Significance:** 3
**Recommendation:** 5

**Clarity, Quality, Novelty And Reproducibility:**

# Clarity
The paper is mostly clear, although there are still some minor points that need to be improved:
- The text of Fig. 1 is too small.
- In section 3.2.1, the authors use lowercase “l” to denote label, which is easy to be confused with the number 1 and the loss “l” (which is also in lowercase).
- Typo in the last paragraph of section 3.2: missing a double quote after “an image of {class label}”.
- Typo in the last line of section 3.2.1: redundant word “the”.
- Typo in the first sentence of the third paragraph of section 4.2: “vanillar”.

# Quality
The quality of the paper has room for improvements since some parts of the paper seem to be written in a hurry.
For example,
- The quality of Fig. 1 is not very good.
- The choice of math notations and the formatting are somewhat casual.
- As an empirical paper, the experiments can be more comprehensive.

# Novelty
The paper presents several novel ideas which are interesting and worth exploring.


**Strength And Weaknesses:**

# Strengths
- The paper studies an important open problem: how to measure model robustness in a general out-of-distribution setting.
- The idea of generating hard test examples for each individual model during evaluation and measuring the robustness of a model with respect to an oracle model is novel.
- The paper is overall well-written and easy-to-understand.

# Weaknesses
- In section 2.2, the authors suggest that the dynamic generation process helps avoid the selection bias of the models. However, how much of the selection bias is avoided is unclear. Actually, it is arguable that current models are affected by the selection bias to a degree that requires any special treatment (see [1]). Besides, the choice of the generative model and the oracle model can also introduce selection bias, albeit less direct than fixed benchmarks. So, the proposed protocol is at best “mitigating” the issue rather than “avoiding” as stated in the paper. This weakens the significance of the paper.
- Throughout all the experiments on ImageNet, the validation rate (VR) is almost the same for models of different architectures and data augmentation techniques. This begs the question whether the generated examples are sufficiently diverse so that they really expose different problems of each model. The provided examples show some visual variations but are not conclusive. Imagine that a model is much more robust than the other, then intuitively, the VR of the robust model should be lower than the VR of the other model because the generated examples for the robust model should be harder for the oracle model. But this is not the case for the models considered in the paper. Moreover, if the oracle model (e.g., CLIP) itself has some weaknesses, then a model that has the same weaknesses would probably have higher OOR than another model that does not share those weaknesses, given that their VR is the same. The paper does not mention if the OOR is sensitive to the choice of the oracle model. All these points are currently unclear and require further investigation.
- Regarding the sparse submodel of VQGAN, the authors first use VQGAN to generate a style-transferred dataset (the first sentence of the third paragraph of section 3.2.1) which is then utilized to find out the feature dimensions related with style. This appears to be circular reasoning. If the style dimensions are unknown and need to be find out, how did the VQGAN generate a style-transferred dataset in the first place? What if there are also significant semantic changes? In that case, the pruned dimensions may also contain many relevent style dimensions. The paper reports that 99.31% of the dimensions are pruned on average, but it only leads to 28.5% runtime reduction on ImageNet. It is not clear if the sparse submodel is efficient enough considering the expense of possible reduction in the diversity of the generated examples.
[1] Recht et al. Do ImageNet Classifiers Generalize to ImageNet?


**Summary Of The Paper:**

This paper questions the validity of fixed benchmarks in evaluating model robustness for real-world applications where the i.i.d. assumption may not hold. Instead, the authors propose to use dynamically perturbed test examples to evaluate each model separately. The perturbed examples must meet two requirements. First, they can still be correctly classified by an oracle model so that most of the underlying causal structure of the original examples are preserved. Second, the examples are perturbed in an iterative manner that maximizes the loss induced by these examples for each given model. A new metric, namely oracle-oriented robustness (OOR), is also proposed for this new evaluation protocol. The OOR of a model is defined as the accuracy of that model on the perturbed examples over the accuracy on the original test examples. With this new evaluation protocol, experiments on ImageNet suggest that strong data augmentation is a key indicator of high OOR despite the model architecture (CNN or ViT). In particular, dynamic data augmentation usually leads to higher OOR than pre-augmented data. Finally, another main empirical finding reported in the paper is that the choice of the generative model used for image perturbation does not affect OOR very much.

**Summary Of The Review:**

The paper studies an important and general problem, and the approach is novel.
Nevertheless, there are some weaknesses to the central arguments of the paper, and the overall quality of the paper just barely meets ICLR standards (which is quite high, in my opinion).
At least some of the issues need to be addressed before I can recommend acceptance.

---

> ### Author Response · Authors · 2022-11-19
> **Response to Reviewer 3rws**
>
> We sincerely appreciate your kind comments and your positive assessment. We hope our point-to-point response can address your concerns.
>
> ### Response to the Proposed Weaknesses
>
> **W1. About the selection bias**
>
> Yes, we totally agree with you that it is inevitable to introduce selection bias when we select specific generators and oracles. The claim here should be revised to "mitigating" rather than "avoiding". With our dynamic generation process, we manage to alleviate such biases. Experiments on different choices of generators show the consistent conclusions made in the paper, which also verifies that we mitigate such biases to some extent. We have updated it in the revised version.
>
> **W2.1. Regarding validation rate (VR) is almost the same for different model architectures**
>
> We are sorry about the confusion. We attribute this phenomenon to the limited computing budget and relatively large perturbation step size.  We conduct our experiment on an NVIDIA GeForce RTX 3090 GPU. With the limited computing resources, we have to set a relatively large perturbation step size in order to reach a tradeoff between image quality and efficiency. In this case, the images are harder to maintain their causal structures at the beginning due to the large perturbation step size, leading to almost the same or only slight fluctuations in VR of different model architectures. If we have more computing resources, we can use a smaller perturbation step size to obtain fine-grained counterfactual images, and observe the corresponding phenomenon reflected in VR.
>
> **W2.2. If the OOR is sensitive to the choice of the oracle model**
>
> That is a good point, and thank you for pointing it out! Indeed, OOR is sensitive to the choice of the oracle model in nature. However, in practice, in addition to finding more powerful oracles with no weaknesses, we still have ways to alleviate such sensitivity. With sufficient computing resources, we can set a relatively smaller stepsize to perturb the image step by step and obtain the fine-grained perturbed images. Admittedly, images corresponding to oracle's weaknesses would still be easier than other images. However, considering the huge performance gap between oracle and the evaluated models, images that are easy for oracle are hard enough for the evaluated models, which can alleviate the sensitivity to some extent and is enough for the applications in reality.
>
> **W3. Regarding issues of sparse VQGAN**
>
> We are sorry for the confusion. The style-transferred dataset is generated based on the original dataset and vanilla VQGAN. To find out the feature dimensions related to style, we utilize LASSO to classify the original images and style-transferred ones and obtain the sparse coefficient matrix of features that LASSO uses when making judgments. We kindly argue the preserved dimensions are highly correlated to VQGAN perturbation and regard them as the style dimensions. The style dimension identification process is not rigorous as expected, as the style dimensions differ between different datasets and parameter configurations of VQGAN. However, they are mostly overlapping, we take the upper bound of the union of different experimental results as the final style dimensions. Admittedly, the pruned dimensions may also contain relevant style dimensions and thus sacrifice the diversity of the generated examples. To solve these issues, we can deploy our method using vanilla VQGAN with sufficient computing resources. Along this way, we can obtain significant runtime reduction without sacrificing the diversity of the generated samples. We leave it as our future work.
>
> ### Revision of Paper
>
> We sincerely appreciate your suggestions and fixed the typos in the revised paper.

---

> > ### Comment · Reviewer_3rws · 2022-11-27
> > **Response**
> >
> > Thanks for your response, but I have to say that my main concerns are not fully addressed. After reading the response, it is still not clear to me whether the proposed evaluation method is currently a better alternative to fixed benchmarks. I feel that more experiments are needed to clearly show this point. For example, experiments showing that fixed benchmarks indeed do not reflect the OOD robustness of models in general, and experiments on different oracle models.

---

### Official Review · Reviewer_QFrF · 2022-11-01

**Confidence:** 4
**Correctness:** 3
**Technical Novelty And Significance:** 3
**Empirical Novelty And Significance:** 3
**Recommendation:** 5

**Clarity, Quality, Novelty And Reproducibility:**

This work is original to my knowledge and there are no major issues with the writing.

Typo: See section 4.4, bullet 3 “robsutness”


**Strength And Weaknesses:**

Strengths:
1. Overfitting benchmarks and evaluating robustness are important problems to the community.
2.  The authors have used a large selection of classification models to test their hypothesis.
3. Comprehensive ablation studies on the oracle and the generator.
4. The paper is fairly simple to understand and concepts are explained well.

Weaknesses:
1. The authors introduce a scoring function $\alpha$ in their image generation algorithm which guides the generation process but there is no concrete example given about what it is. I’m assuming this scoring function (at least the one used in the paper) essentially maximizes the loss of a given image in the direction of a different class. This is the same as Santurkar et al. where they use the inner maximization problem to generate images using a classifier.
2. I was under the impression that OOR values would be <1 since standard accuracy would usually be higher than counterfactual accuracy but all values in the paper seem to be > 10 at least. Is there a ‘*100’ missing?
3. The authors claim that images generated using their method are diverse. However, no diversity metric are given. Since there is a computational budget, a lack of diversity of images could artificially inflate the robustness of a model.
4. Is OOR supposed to give a ranking of robust models? I ask because in Table 6 in Appendix D, a vanilla model has higher OOR than PGD (so more robust than PGD). I find this hard to believe. This must mean that OOR specifically depend on the threat model being used. The authors should be more explicit about this limitation.
5. It is unclear how OOR behaves when you have different generators but the same oracle.
6. Since we are using the oracle, any gaps in the robustness of the oracle will also show up in the model being evaluated. The authors, to their credit, mention this in Section 5 but it is still a major limitation.

**Summary Of The Paper:**

This paper introduces an evaluation protocol to study robustness of image classification models. The protocol essentially creates a new test set depending on the model being evaluated and the dataset it was trained on. This, the authors argue, would reduce dependence on static benchmarks and create a “dynamic evaluation protocol”. However, this creates an issue where robustness of models cannot be directly compared since they no longer have the same test set. An oracle (CLIP) is introduced which helps us compare robustness fairly.

The authors provide an algorithm to generate new, ‘perturbed’ images. We use a generic generator model (VQGAN in the paper) to perturb the clean image, then check whether this perturbation has the same label as the original label (using the oracle). This step is repeated several times to stay within a budget.

The above algorithm applied to images in the clean dataset produces perturbed images which act as a test set. This test set depends on the choice of the model being evaluated, oracle and generator. Therefore, it is a general, dynamic way of evaluating robustness.
The authors also introduce a new metric called oracle-oriented robustness (OOR) = (Counterfactual acc)/(standard acc). This is used to measure the robustness gap to the oracle.

MNIST, CIFAR-10 and ImageNet are used for experiments with several vanilla image classification models (ResNet etc) and some models with claimed robustness properties (AugMix, DeepAugment etc).

**Summary Of The Review:**

Overall, I think the authors have proposed a fairly simple and solid idea with good evaluation, however, there seem to major limitations about its applicability which need to be clarified. I think this is just below the publication threshold for me.

---

> ### Author Response · Authors · 2022-11-19
> **Response to Reviewer QFrF**
>
> We sincerely appreciate your kind comments and your insightful suggestions. Regarding the weaknesses, we hope our point-to-point response can address your concerns.
>
> **W1. Regarding details of the scoring function $\alpha$**
>
> We are sorry we missed the details of the scoring function, and thank you for pointing it out! The scoring function essentially maximizes the classification loss of the evaluated model to guide the generation of images. We have incorporated the details in the updated draft. Please kindly check the blue text in the method section.
>
> **W2. Is there a ‘*100’ missing for OOR?**
>
> We are sorry about the confusion, and thank you for pointing it out! Yes, there should be a '*100' for the OOR value. We have updated the formula of OOR in the updated version. Please kindly check the blue text in Section 3.3.
>
> **W3. Regarding diversity issues.**
>
> We are sorry about the confusion, and thank you for pointing it out! We think the claim made here is not appropriate. We use the scoring function that maximizes the classification loss of the evaluated model to guide the generation of images, which is not directly related to generating diverse images, but would lead to generating adversarially effective ones. We kindly argue that generating effective (hard) images, rather than diversified but superficial (easy) ones is more beneficial to expose defects of evaluated models, which is what the evaluation protocol values. Therefore, a more reasonable claim should be the images generated using our method are effective to expose the models' weaknesses. Please kindly note that our method is able to dynamically adjust attack strategies based on different model properties, and automatically generate effective counterfactual images (See Figure 2).
>
> **W4. A vanilla model has a higher OOR than PGD (81.73 v.s. 66.18)**
>
> This is a good point! In Table 6 in Appendix D, we find that the adversarially trained model and vanillaly trained model process the data differently. As our method would dynamically adjust attack strategies based on different model properties, the transferability of the generated images would barely hold if the models' properties differ significantly. For example, the counterfactual images generated for PGD model would be too easy for the vanilla model, and thus, the OOR becomes meaningless. However, such cases only happen in the static evaluation process where the generation of the test images for vanilla models is static, rather than dynamically generated based on vanilla models' properties. Therefore, we kindly argue that OOR should be used with our dynamic evaluation protocol, i.e., the OOR of Van. model tested with the images generated by the vanilla model itself **(57.79)** is lower than the OOR of PGD model tested with the images generated by the PGD model itself **(66.18)**.
>
> **W5. How OOR behaves when you have different generators but the same oracle**
>
> We find that the conclusion is consistent under different generator choices. Please kindly check Section 4.5 for more details. We hope this may ease your concern about different generators.
>
> **W6. Limitations of the oracle.**
>
> This is a good point! We have to admit that this is the major limitation. However, in addition to looking for a more powerful oracle, we still have ways to alleviate this limitation by increasing the computing resources. In this paper, we conduct our experiment on an NVIDIA GeForce RTX 3090 GPU. With the limited computing resources, we have to set a relatively large perturbation step size in order to reach a tradeoff between image quality and efficiency. Therefore, our validation rate is usually only around 30%, and the images about chimpanzees are often relatively easy due to the large perturbation step size. If we have sufficient computing resources, we can set a relatively smaller stepsize to perturb the image little by little and can get enough more perturbed images. Admittedly, the oracle's bias still exists here, that is, the chimpanzee images are still easier than other ones. However, considering the huge performance gap between oracle and the evaluated models, images that are easy for oracle are hard enough for the evaluated models, which is enough for the applications in reality. We leave the deployment on sufficient computing resources as a future work.
>
> **Revision of Paper**
>
> We sincerely appreciate your suggestions and fixed the typos in the revised paper.

---

### Decision · Program_Chairs · 2023-01-20

**Decision:**

Reject

**Justification For Why Not Higher Score:**

According to my expertise and reviewing process, this paper should belong to a Reject.

**Justification For Why Not Lower Score:**

According to my expertise and reviewing process, this paper should belong to a Reject.

**Metareview: Summary, Strengths And Weaknesses:**

This paper proposes an adversarial attack benchmark, where the attacked samples are generated by high-quality generative models filtered by a "surrogated oracle" (a model trained with large-scale extra data points, such as CLIP). More specifically, the proposed method generates adversarial samples by maximizing classification loss (i.e., cross-entropy loss) using VQGAN. The generated image is validated by the CLIP model, and if the CLIP model predicts the generated image with a wrong label, then the generated image is set to the original image. With the generated adversarial samples, this paper shows that the existing vision models are not robust to the proposed attack method.

However, there are several obvious weakness: 1) This paper has very limited novelty and the quality of this paper has a large room for improvement. 2) The proposed method highly depends on VQGAN and CLIP models. It will cause serious problems: namely, the generation quality itself is limited by VQGAN and CLIP performances, and the metric is biased toward the CLIP zero-shot performance. Hence, measuring the proposed benchmark will measure how a model is similar to the CLIP model, not measure how a model is robust. 3) Whether the proposed evaluation method is currently a better alternative to fixed benchmarks. More experiments are needed to clearly show this point. For example, experiments showing that fixed benchmarks indeed do not reflect the OOD robustness of models in general, and experiments on different oracle models. Overall, this paper may not be ready for publication at ICLR. The next version must be a strong paper if authors can take comments into consideration.